# Quantifying compartment-associated variations of protein abundance in proteomics data

Luca Parca[1,†], Martin Beck[1,2] (iD), Peer Bork[1,3] (iD) & Alessandro Ori[4,*] (iD)

## Abstract

Quantitative mass spectrometry enables to monitor the abundance of thousands of proteins across biological conditions. Currently, most data analysis approaches rely on the assumption that the majority of the observed proteins remain unchanged across compared samples. Thus, gross morphological differences between cell states, deriving from, e.g., differences in size or number of organelles, are often not taken into account. Here, we analyzed multiple published datasets and frequently observed that proteins associated with a particular cellular compartment collectively increase or decrease in their abundance between conditions tested. We show that such effects, arising from underlying morphological differences, can skew the outcome of differential expression analysis. We propose a method to detect and normalize morphological effects underlying proteomics data. We demonstrate the applicability of our method to different datasets and biological questions including the analysis of sub-cellular proteomes in the context of *Caenorhabditis elegans* aging. Our method provides a complementary perspective to classical differential expression analysis and enables to uncouple overall abundance changes from stoichiometric variations within defined group of proteins.

**Keywords** cellular compartment; differential expression; linear model; organelle; proteomics
**Subject Categories** Genome-Scale & Integrative Biology; Methods & Resources; Post-translational Modifications, Proteolysis & Proteomics
**Mol Syst Biol. (2018) 14: e8131**

## Introduction

Mass spectrometry-based proteomics has been successfully used to determine sub-cellular protein composition, discover new portions of the cellular interactome, and map post-translational modifications. Different experimental strategies have been developed to perform quantitative experiments where differences in protein abundance are determined, for example, by introducing stable isotopes in one of the experimental conditions tested (Ong *et al*, 2002; Ong & Mann, 2005). A major factor for the interpretation of the experimental outcome is data processing (Park *et al*, 2003). The main data processing strategies used in proteomics, e.g., scaling by mean or median through quantile normalization, have been developed for microarray and RNAseq data, with the underlying assumption that total level of mRNA in the cell is stable and does not differ significantly between the compared samples (Bolstad *et al*, 2003). For transcriptomics data, it has been shown that global changes in transcript levels, e.g., down-regulation of all transcripts due to general inhibition of transcription, can introduce artifacts when standard differential analysis approaches are employed (apparent up- and down-regulation of transcripts instead of, e.g., widespread down-regulation; Jaksik *et al*, 2015). Similarly, profound morphological differences between cellular types or states can also influence the outcome of comparative genomic and proteomic analysis (Lin *et al*, 2012; Lovén *et al*, 2013). Lundberg *et al* (2008) showed that the majority of all proteins are expressed in a cell size-dependent fashion and that the comparative analysis of the protein expression values requires a normalization procedure. Additionally, it is known that different tissues show different levels of respiratory activity and variable amounts of mitochondria (Kirby *et al*, 2007; Fernández-Vizarra *et al*, 2011) and that the number and composition of organelles can be affected, for example, by the aging process (Cellerino & Ori, 2017). Covariation of protein abundance across different conditions can be also exploited, and it can contribute to functional proteomics (Kustatscher *et al*, 2016). Currently, there is a lack of systematic approaches able to detect and deal with differences in cellular organization that might influence the outcome of proteomics data analysis from unfractionated samples.

Here, we show that in several published proteomics datasets proteins associated with different cell compartments/organelles show distinct distribution of fold changes across the conditions tested. This manifests as a consequence of underlying differences in cell morphology that are not taken into account by classical differential expression tools. Although such differences provide robust signals about different cell states and, as such, can be used as

1  Structural and Computational Biology Unit, European Molecular Biology Laboratory, Heidelberg, Germany
2  Cell Biology and Biophysics Unit, European Molecular Biology Laboratory, Heidelberg, Germany
3  Max-Delbrück-Centre for Molecular Medicine, Berlin, Germany
4  Leibniz Institute on Aging-Fritz Lipmann Institute, Jena, Germany
   *Corresponding author. Tel: +49 3641 65 6808; E-mail: alessandro.ori@leibniz-fli.de
   †Present address: Department of Biology, University of Rome "Tor Vergata", Rome, Italy

biomarkers, the non-uniform distribution of fold changes can mask biologically relevant alterations in the composition of cell compartments/organelles. We thus propose an approach that is able to better reflect compartment-specific protein changes (Fig 1A), and we experimentally validate this by analyzing the proteomic changes identified in nuclei isolated from different cancer cell lines compared to their total lysate. Using our approach, changes in protein abundance identified in comparative proteomic studies can be re-interpreted to better reflect the context of protein sub-cellular localization, and to provide an additional level of detail about the biological differences between cellular states. To demonstrate this, we re-analyzed a dataset of chronological aging in the nematode *Caenorhabditis elegans* and observed heterogeneous abundance changes among mitochondrial and extracellular proteins implying an age-dependent remodeling of these cellular compartments. These novel biological insights were not apparent when traditional analysis methods were applied. Our approach is broadly applicable to large-scale proteomic studies, and we anticipate analogous strategies to be derived for different context levels such as protein complexes and pathways.

## Results and Discussion

### Compartment-specific shifts in protein abundance are apparent in large-scale proteomics dataset

We analyzed seven mass spectrometry datasets covering the proteomes of different mammalian tissues (Geiger *et al*, 2013), cell types (Azimifar *et al*, 2014; Sharma *et al*, 2015), healthy and diseased states (Wiśniewski *et al*, 2012; Guo *et al*, 2015; Tyanova *et al*, 2016), and cancer development stages (Wiśniewski *et al*, 2015). In these experiments, the abundance fold change (FC) of thousands of proteins has been calculated using standard differential analysis approaches (see Materials and Methods section). To investigate whether changes in organelle number or size are reflected in these datasets, we assigned cellular localization to each of the quantified proteins using Gene Ontology (GO) term annotation (The Gene Ontology Consortium, 2015). On average, 78% of the proteins in the analyzed dataset could be annotated using GO cellular compartments terms (Fig EV1A). We compared abundance changes of proteins belonging to four major cellular compartments (nucleus, cytoplasm, mitochondrion, and extracellular space) which, on average among the analyzed datasets, accounted for 96% of all the annotated proteins. In all the seven datasets, we observed that proteins assigned to specific cellular compartments tend to display similar protein fold changes, indicating that their abundances are associated with each other. We therefore calculated protein fold change distributions for each cellular compartment and found statistically significant shifts between such distributions (Fig 1B–D). We found a distinct increase in mitochondrial proteins in liver cells that can be readily captured from the comparison of lung and liver tissue proteomes (Geiger *et al*, 2013; average change of +1 $\log_2$FC, Mann–Whitney test $P = 5.3 \times 10^{-32}$; Fig 1B and Dataset EV1), consistent with the knowledge that hepatocytes have an elevated number of mitochondria as compared to other cell types (Veltri & Espiritu, 1990). Similar differences can be also detected between more closely related cell types deriving from the same

organ. For instance, we found increased abundance of nuclear and extracellular proteins and decreased abundance of mitochondrial proteins in Kupffer cells vs. hepatocytes (Azimifar *et al*, 2014; average change of +0.4, +0.3 and −1.2 $\log_2$FC, Mann–Whitney test $P = 3.3 \times 10^{-22}$, $P = 8.5 \times 10^{-03}$, and $P = 1.9 \times 10^{-88}$, respectively; Fig 1C and Dataset EV2).

Major morphological changes can also be a consequence of disease such as malignant transformation. We analyzed the abundance of proteins in healthy kidney cells and renal carcinoma cells (Guo *et al*, 2015) and found a decrease in mitochondrial proteins in cancer cells (average change of −0.5 $\log_2$FC, Mann–Whitney test $P = 1.8 \times 10^{-22}$; Fig 1D and Dataset EV3). In addition, we also observed progressive shifts in the relative abundance of nuclear (Mann–Whitney test $P < 0.01$) and extracellular (Mann–Whitney test $P < 0.01$) proteins between healthy colorectal mucosa, adenomas, and colon cancers (Wiśniewski *et al*, 2015; Fig EV2). Subsequently, we extended the analysis to proteins mapping to six additional organelles: endoplasmic reticulum, Golgi apparatus, cell membrane, nuclear membrane, lysosome, and peroxisome (Fig EV1C–E). This allowed us to observe a previously unappreciated correlation between the abundance changes of proteins annotated as peroxisomal and mitochondrial (Pearson's $R = 0.97$, $P = 3 \times 10^{-04}$) that manifested in all the seven different datasets used (Fig EV1F).

Collectively, our analysis indicates the widespread existence of cell compartment-specific shifts in the output of comparative mass spectrometry experiments reflecting morphological differences between the compared cell states. These major shifts can be detected using protein annotation and a simple statistical test and, if present, should be taken into account when interpreting the data. However, this approach does not inform about variations of protein abundance within the same cellular compartment. As an example, a mitochondrial protein complex might appear increased in abundance, reflecting an increase in mitochondrial number or size, although its actual abundance with respect to all other mitochondrial proteins remains unchanged.

### Differential protein expression analysis in the context of cell compartments

In order to gain insight into the composition of cellular compartments across cell states, we propose a normalization approach that complements standard differential analysis by taking into account differences in size or abundance of cell compartments. Our approach aims at partitioning total proteome data using prior knowledge deriving from the GO annotation, and calculates new relative abundances for proteins belonging to major cellular compartments. For each compartment, a linear model is built from the abundances of proteins in the two conditions compared (Fig 2A). In all the datasets that we tested, the $\log_2$ abundances of proteins annotated to same cell compartment followed linear models between the compared samples; therefore, non-linear modeling was not explored (see Materials and Methods section). Each linear model was evaluated through its statistics, namely the $P$-value and the $R^2$ (Dataset EV4A). In each linear model, the distance, i.e., the residual value, between the protein abundance and the linear fit, can be used as a compartment-normalized variation (CNV) value. This value reflects the relative abundance difference of a protein compared to

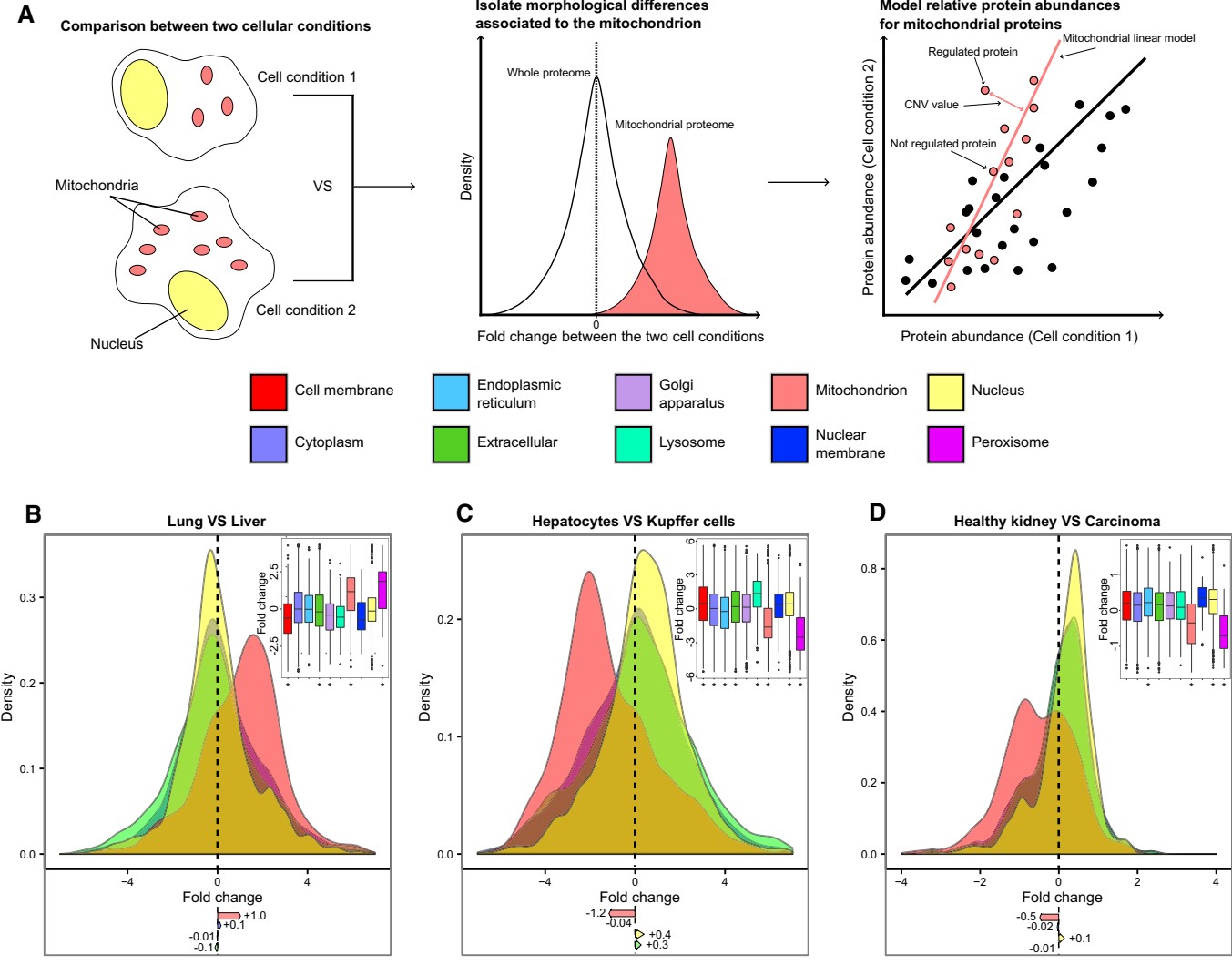

**Figure 1.  Differences in cellular organization emerge from quantitative proteomic experiments.**

A       Scheme of the analysis of cellular compartments based on modeling compartment-normalized variation (CNV) values for proteins (shown for two hypothetical cell conditions displaying different numbers of mitochondria).

B–D   Analysis of protein abundance shift in cellular compartments in published datasets. Density plots for protein fold change distributions in different compartments for (B) lung vs. liver cells (Geiger *et al*, 2013), (C) hepatocytes vs. Kupffer cells (Azimifar *et al*, 2014), and (D) healthy human kidney vs. renal carcinoma cells (Guo *et al*, 2015). Distribution of protein fold changes for 10 different compartments is shown as boxplot (inset); mean fold changes (log$_2$) are indicated below the density plots; below the boxplot, asterisks mark the cellular compartments that show significantly different distribution of fold changes compared to the whole proteome (Mann–Whitney test *$P <$ 0.01).

Data information: Boxplots: the horizontal line represents the median of the distribution, the upper and lower limit of the box indicate the first and third quartile, respectively, and whiskers extend 1.5 times the interquartile range from the limits of the box. Values outside this range are indicated as outlier points. Related to Figs EV1 and EV2 and Datasets EV1–EV3.

its cellular compartment (Fig 2B and Dataset EV3). Since many proteins are associated with more than one cellular compartment (on average, only 20% of the annotated proteins were specific to one compartment and 36% were annotated to two compartments, Fig EV1B; Thul *et al*, 2017), we wanted to assess the robustness of the linear models when taking into account multiple compartment annotations for the same proteins. Thus, we compared the CNV models built using all the proteins mapping to a given compartment to CNV models built using only proteins that are exclusive to a given compartment, so that there are no shared proteins between

the linear models. We measured an average Pearson correlation of 0.97 between the CNV values for the same proteins using the two types of models. In the case of mitochondria, we evaluated an independent and curated annotation of mitochondrial proteins from MitoCarta (Calvo *et al*, 2016), and used it to build the mitochondrial-exclusive CNV model. The average Pearson correlation between the previous models and these mitochondrion-exclusive models was 0.99. The statistics of the CNV models and their correlation with compartment-exclusive models are reported in the Dataset EV4A. Finally, we tested whether proteins belonging to different

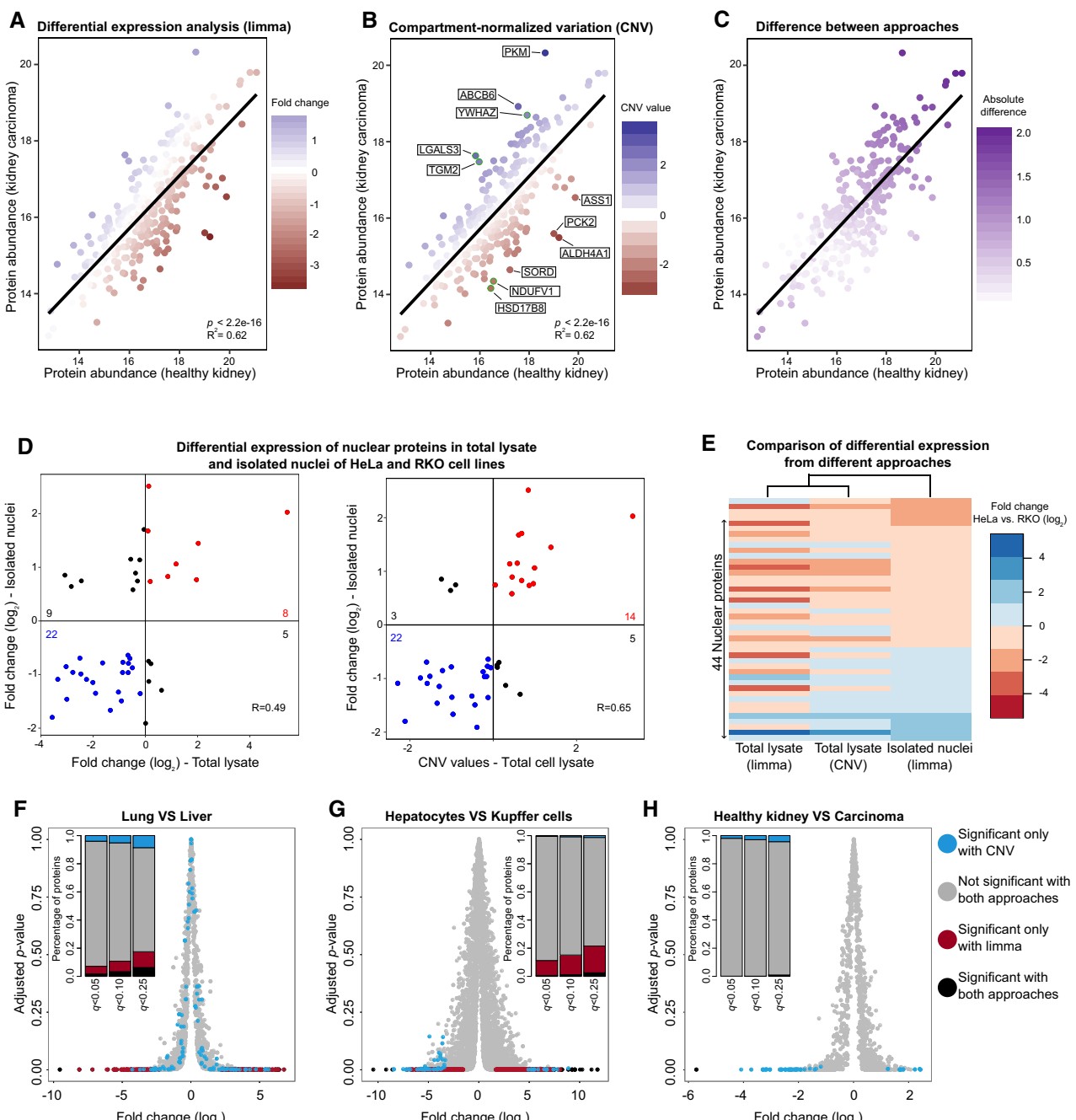

**Figure 2. Compartment-specific analysis reveals differences in organelle composition that can be validated by sub-cellular fractionation.**

A–C  Mitochondrial proteins are plotted using their absolute abundance (IBAQ score) in healthy kidney vs. renal carcinoma cells (Guo *et al*, 2015). Each mitochondrial protein is colored according to (A) its fold change calculated by standard differential expression using the limma package (Ritchie *et al*, 2015; Phipson *et al*, 2016); (B) its CNV value (five proteins with the highest CNV value and five proteins with the lowest CNV values are highlighted and annotated in boxes); and (C) the absolute difference between the two values.

D  Correlation between standard fold change values (left panel) and CNV values (right panel) of proteins quantified in the total lysate and isolated nuclei of HeLa and RKO cells. Only proteins that are differentially regulated (adj. $P < 0.05$) in isolated nuclei are shown.

E  Clustering of nuclear protein fold changes estimated from whole cells and isolated nuclei, and comparison to the CNV values obtained from whole cell data.

F–H  Comparison of standard differential expression and CNV approach for the three datasets shown in Fig 1B–D. Volcano plots based on fold changes and adjusted *P*-values (limma) obtained for (F) lung vs. liver cells (Geiger *et al*, 2013), (G) hepatocytes vs. Kupffer cells (Azimifar *et al*, 2014), and (H) healthy kidney vs. renal carcinoma cells (Guo *et al*, 2015). Proteins are colored depending on their significance when using the standard limma approach and the CNV approach (based on the four main compartments, *q*-value < 0.1). A stacked barplot (inset) shows the percentage of unique proteins belonging to each category for three *q*-value thresholds (0.05, 0.1, and 0.25).

Data information: Related to Figs EV3 and EV4, and Datasets EV1, EV2, EV3, and EV5.

compartments are more likely to be detected as significantly affected by the CNV approach. We did not observe a significant association between multiple compartment annotation for proteins and their classification as differentially expressed by the CNV approach (Dataset EV4B). Thus, we conclude that the linear models underpinning the CNV approach are robust regardless of whether proteins with multiple compartments are considered or not.

Depending on the extent of the cellular compartment shift in whole proteome data, proteins can be assigned different standard fold change and CNV values (Fig 2C). Therefore, in order to evaluate the performance of the CNV approach, we analyzed the proteome profiles obtained from two cell lines that are known to have distinct morphological features. The commonly used HeLa cells and the colon carcinoma-derived cells (RKO) have drastically different morphology and nuclear size, due to a > 2-fold smaller RKO nuclear size as compared to HeLa (average nuclear surface area 206.3 and 543.5 Å$^2$ for RKO and HeLa, respectively; Fig EV3). Proteome profiles from whole cell extract and isolated nuclei are available for both cell lines (Geiger *et al*, 2012; Ori *et al*, 2013). Indeed, when we analyzed whole cell data using a standard differential expression analysis tool [limma package (Ritchie *et al*, 2015; Phipson *et al*, 2016)], we observed that the majority (70%) of the nuclear proteins differentially regulated in isolated nuclei are classified as down-regulated in RKO cells compared to HeLa cells, reflecting the smaller nuclear size of the former cell line. However, the same analysis performed on data from isolated nuclei showed that ratio between up- and under-regulated proteins is, as expected, more balanced, reflecting differences in the composition of the nucleus between the two cell lines. The discrepancies between the fold changes estimated from whole cell and isolated nuclei result in a significant, but modest, correlation between the two datasets ($R = 0.49$; Fig 2D, left panel). We then re-analyzed the whole cell data treating cellular localizations independently using our CNV approach (Dataset EV5). We found that our approach is able to take into account the differences attributed to changed morphology and provides fold change values for nuclear proteins that are considerably closer to the values obtained from isolated nuclei, with an improved correlation ($R = 0.65$) between whole cell and isolated nuclei estimates (Figs 2D, right panel, and E). These data show that our CNV approach is useful to derive insights on the proteome of a cell compartment when applied to total proteome data, irrespectively of morphological differences between the compared samples.

## Comparison of proteome-wide and compartment-specific differential expression analysis

In order to quantify the impact of the CNV approach on the analysis of proteomics data, we compared the outcome of standard differential expression and CNV approach for the three datasets where we detected significant compartment shifts (Fig 1B–D). Direct comparison of the statistics revealed low correlation between *q*-values assigned by the two approaches (Fig EV4), indicating complementarity between them. Notably, the extent of complementarity was not uniform between datasets, being more pronounced when different tissues (e.g., liver vs. lung) are compared (Fig 2F and H). We reasoned that the CNV can provide two additional levels of information: (i) It can reveal alterations of protein level that reflect a

compartment-wide abundance change rather than a protein-specific one; and (ii) it can discover new protein changes that emerge only after normalizing for compartment-wide changes. Therefore, we explicitly investigated the overlap between significant proteins identified by standard differential expression and CNV approach. Across the three datasets tested, we found a variable proportion of cases (ranging between 50 and 92%) that were identified as differentially expressed by the standard approach (*q*-value < 0.1), but are very close to the linear model of their respective compartment, and, thus, classified as not significant with the CNV approach (Fig 2F–H, colored in red). We interpret these cases as deriving from compartment-wide abundance changes. This effect was particularly pronounced for the Azimifar *et al* (2014) dataset that showed very prominent shifts for nuclear, mitochondrial, and extracellular proteins (Fig 1C). Regarding newly discovered cases, we found 104, 53, and 38 proteins that were identified as significant (*q*-value < 0.1) exclusively by the CNV approach, respectively, for the lung vs. liver (Geiger *et al*, 2013), hepatocytes vs. Kupffer cells (Azimifar *et al*, 2014), and healthy kidney vs. carcinoma (Guo *et al*, 2015) datasets (Fig 2F–H, colored in cyan). The majority of these cases display low fold changes relative to the total proteome, but appears as outlier in the linear models for the respective compartment. Taken together, these data demonstrate that protein expression can be analyzed in the context of cellular compartment by building simple linear models that allow a complementary interpretation of the results of canonical differential expression (Fig EV4), revealing new differences in the abundance of proteins belonging to the same compartment across cell types and states.

## Reinterpreting age-related changes of the *C. elegans* proteome

To illustrate how the CNV approach can provide novel biological insights, we analyzed a time-series mass spectrometry experiment related to aging of *C. elegans* (Walther *et al*, 2015). The abundance of proteins was calculated at five different time points during the lifespan of the organism (days 1, 6, 12, 17, and 22) using the SILAC approach (Ong *et al*, 2002). In the original publication, the authors noted that different cellular compartments were differently affected by aging in terms of protein abundance changes. We re-analyzed this dataset using the CNV approach by assigning proteins to four main cellular localizations and investigating the respective SILAC ratios as a function of aging in a compartment-specific manner. We observed a general age-dependent increase in abundance for nuclear and extracellular proteins, while mitochondrial proteins were progressively decreasing in abundance with age (Mann–Whitney test $P < 0.01$, Fig 3A–C). The CNV approach highlighted an age-dependent decrease in members of the cytosolic ribosome and an increase in the proteasome, relative to their respective cell compartment (Mann–Whitney test $P < 0.01$; Fig EV5 and Dataset EV6). Indeed, the same age-related variations were observed by the authors of the original publication, and they are corroborated by other independent studies (Hsu *et al*, 2003; Golden & Melov, 2004; Vilchez *et al*, 2012; Kirstein-miles *et al*, 2013).

The complex I of the respiratory chain was originally described to decrease in abundance during *C. elegans* aging (Fig 3D, left panels), similar to what has been observed for other species (Zahn *et al*, 2006; Ori *et al*, 2015). However, using the CNV approach, we found that the majority of complex I members do not change

significantly relative to the rest of mitochondrial proteins (Fig 3D, right panels, and Dataset EV6). We instead revealed two members of the complex I (the nuclear-encoded NDUF-2.2 and the mitochondrial-encoded ND5) that showed extreme and opposite CNV values, suggesting loss of stoichiometry due to mitonuclear imbalance

(Fig 3E; Houtkooper *et al*, 2013). Similarly, the abundance of mitochondrial ribosome components was suggested to be decreasing with age. However, using the CNV approach, we could show that the abundance of this complex relative to the rest of the mitochondrial proteome appears to be stable (Fig 3D and E). Conversely, we

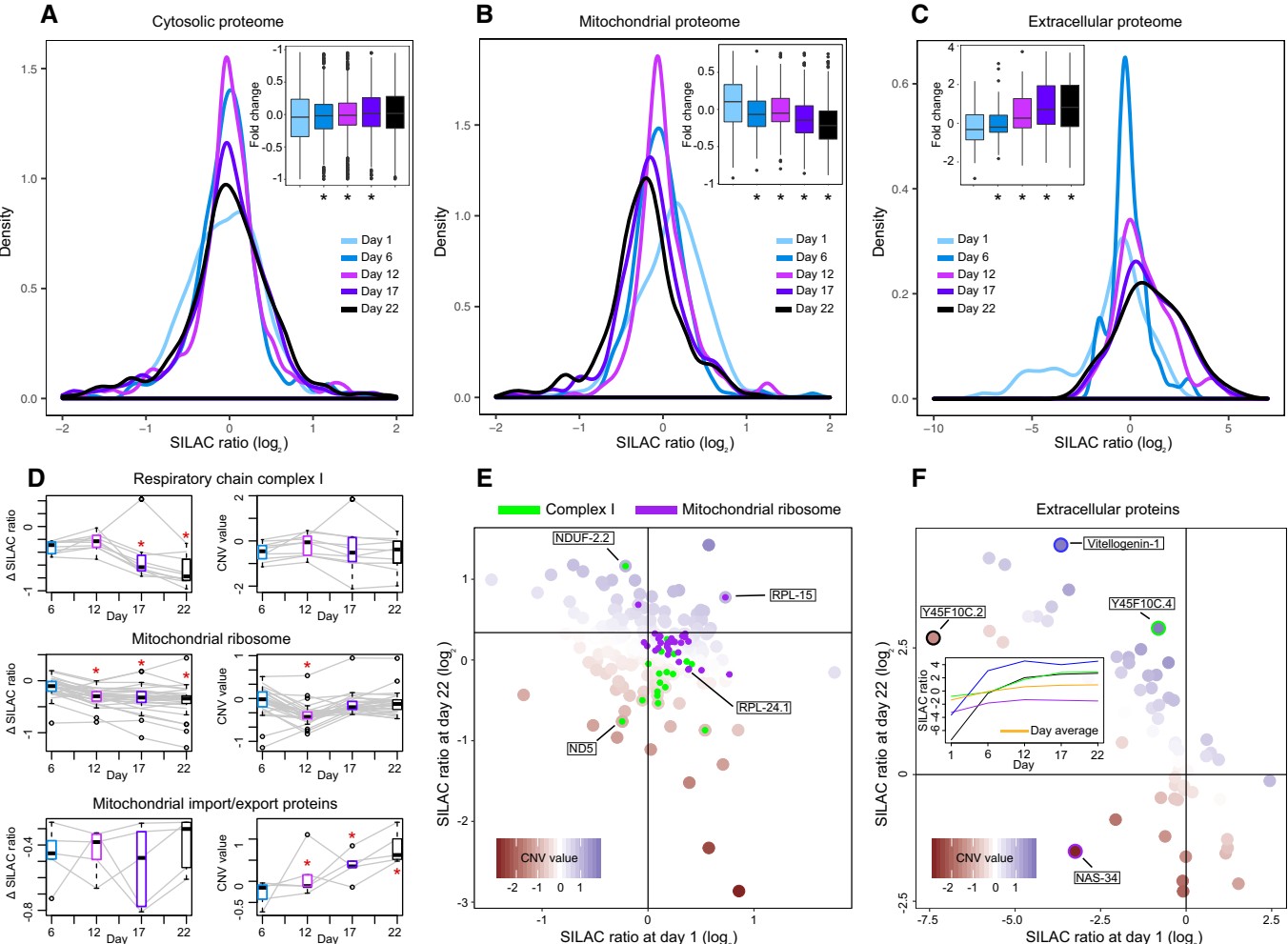

**Figure 3. Remodeling of the mitochondrial and extracellular proteome during *Caenorhabditis elegans* aging.**

A–C   Analysis of the abundance distribution for (A) cytoplasmic, (B) mitochondrial, and (C) extracellular proteins across five age groups in *C. elegans* (Walther *et al*, 2015). Each age group was compared against a pool reference sample using SILAC (Ong *et al*, 2002). Distribution of protein fold changes for the different time points is shown as density plots and as boxplots (inset); below the boxplots, asterisks mark the time points that show significantly different distribution of fold changes compared to the first time point (Mann–Whitney test *$P < 0.01$).

D   Abundance variation of complex members for three mitochondrial complexes (respiratory complex I, mitochondrial ribosome, and mitochondrial import/export proteins). For each complex, the abundance variation of each member (colored in gray) over the five age groups is reported with the $\Delta$ SILAC ratio (difference between the experimental SILAC ratio of a time point and the SILAC ratio at day 1, left) and the CNV value (right). For each age group, the boxplot (colored as the corresponding time point in the panels A–C) represents the variance between the abundances of the complex members. Significant changes (Mann–Whitney test $P < 0.05$) are marked with a red star.

E   Variation of the mitochondrial proteome over the lifespan of *C. elegans* (day 1 and day 22). Respiratory chain complex I and mitochondrial ribosomal proteins are highlighted in green and purple, respectively (selected proteins, with extreme CNV values, are indicated).

F   Variation of the extracellular proteome abundance over the lifespan of *C. elegans* (day 1 and day 22). Four proteins have been selected (highlighted with a colored circle: purple for NAS-34, green for Y45F10C.4, blue for Vitellogenin-1, and black for Y45F10C.2) for their slower or accelerated increase in abundance compared to the rest of extracellular proteins. The variation of the selected proteins (highlighted with the same colors) throughout the worm lifespan is represented in the middle box, with the average of the extracellular proteome colored in yellow.

Data information: Boxplots: the horizontal line represents the median of the distribution, the upper and lower limit of the box indicate the first and third quartile, respectively, and whiskers extend 1.5 times the interquartile range from the limits of the box. Values outside this range are indicated as outlier points. Related to Fig EV5 and Dataset EV6.

observed that proteins forming the inner and outer mitochondrial membrane transport systems increase their abundance with age (Mann–Whitney test $P = 3.9 \times 10^{-3}$; Fig 3D). This phenomenon might be linked to protein turnover imbalances in the mitochondrion and the age-related increase in harmful and damaged proteins in the organelle (Walther *et al*, 2015). These results indicate that the mitochondrial proteome, as a whole, decreases during aging, and the variation in the abundance of specific proteins hints to re-arrangement of mitochondria composition potentially underlying ultrastructural differences (Brandt *et al*, 2017).

The application of the CNV approach to SILAC experiments based on a common reference sample provides an additional level of analysis that is exemplified by the behavior of extracellular proteins during *C. elegans* aging. Extracellular proteins are generally characterized by a significant increase in abundance as *C. elegans* ages, with an average SILAC ratio ($\log_2$), relative to common reference sample, of −1.4 on day 1 and of +1.0 on day 22 (Fig 3C). This is reflected by a linear relationship between SILAC ratios of day 1 and day 22 ($P = 1.6e^{-10}$) with extracellular proteins that show a low SILAC ratio on day 1 showing a high SILAC ratio in old worms (Fig 3F). CNV values can be used to capture differences among extracellular proteins and estimate the relative rate of abundance change with age. Among proteins showing the most extreme CNV values, we observed Vitellogenin-1 (CNV value of +1.5), a precursor of egg-yolk proteins, the Y45F10C.4 protein (CNV value of +2.1), and the UPF0375 protein Y45F10C.2. The latter protein, known to negatively regulate the egg-laying rate (Hao *et al*, 2011), shows a SILAC ratio of −7.4 on day 1 and increases it to +2.7 on day 22. However, its CNV value is low (−1.6), indicating that the abundance of this protein is not increasing at the same rate as other extracellular proteins. Similarly, the Zinc metalloproteinase nas-34 shows a SILAC ratio of −3.2 on day 1 and of −1.5 on day 22, thus increasing its abundance. The CNV approach assigned to it a value of −3, which is the lowest score of the whole extracellular compartment. The different dynamics of extracellular proteins suggest remodeling of the extracellular matrix composition during *C. elegans* aging.

In summary, we show that morphological differences between cell types, states, or conditions are reflected in the abundance of proteins associated with particular cellular compartments. This results in global fold change shifts for proteins associated with different compartments. Such abundance shifts are extremely robust (deriving from tens to hundreds of proteins), and they can be used as markers of cell identity. Currently used data processing approaches do not contemplate such differences that affect collectively large portions of the proteome. This poses limitations to the detection of variations in composition of cellular compartments. The aim of the CNV approach presented here is to integrate standard analysis by revealing compartment-specific changes that can be hidden in whole proteome data. We show that a compartment-based partitioning of proteome data followed by normalization of protein abundances according to a linear model recapitulates variations in protein abundance observed in sub-cellular fractionation experiments, and it can be used to reveal proteome alterations within a cell compartment that might underline, e.g., age-related differences in the morphology of an organelle (Brandt *et al*, 2017). Our approach does not rely on sample fractionation and can be applied to data obtained from whole cell/tissue extracts. However, it depends on compartment assignment of proteins and, therefore, it relies on accurate annotation of protein localization, thus limiting its application to well-annotated model organisms. For this reason, the CNV approach cannot be useful for the annotation of organelle catalogues of proteins. Moreover, the reliability of the linear models built for each compartment depends on the number of proteins used to build them; therefore, the statistics of models built for organelles represented by few proteins must be carefully evaluated. Our concept can, in principle, be extended to other linear and non-linear modeling approaches, and applied to other interdependencies of proteins such as finer grained sub-cellular localizations and structures (Christoforou *et al*, 2016; Mulvey *et al*, 2017), protein complexes, or pathways, to identify true drivers of biological differences between cell types and states.

# Materials and Methods

### Datasets

The following dataset were analysed in this work: (i) SILAC quantification of proteins in mouse lung and mouse brain, each with three replicates (Geiger *et al*, 2013). SILAC ratio was $\log_2$-transformed. (ii) Proteome quantification in murine liver (Azimifar *et al*, 2014). Protein intensities from Kupffer cells, hepatic stellate cells, and hepatocytes (each with four replicates) were collected and $\log_2$-transformed. (iii) Mouse brain proteome resolved by cell type (Sharma *et al*, 2015). Protein intensities from microglia (young and old, each with three replicates), neurons (nine replicates), and astrocyte (three replicates) were collected and $\log_2$-transformed. (iv) Proteome quantification in breast cancer subtypes (Tyanova *et al*, 2016). Normalized SILAC ratios of proteins from triple negative (11 replicates), Her2-positive (15 replicates), and ERPR (14 replicates)-positive breast cancer were collected. (v) Protein quantification of nine paired (healthy/cancer) kidney biopsies (Guo *et al*, 2015), each with two replicates. One of the nine patients was removed from the analysis as the distribution of the protein abundances was not consistent with other samples. Protein intensities were $\log_2$-transformed. (vi) Proteome quantification of normal colon tissue (eight samples) and adenocarcinoma (eight samples; Wiśniewski *et al*, 2012). Protein intensities were $\log_2$-transformed. (vii) Proteome analysis of colorectal mucosa (eight samples), adenoma (16 samples), and cancer (eight samples; Wiśniewski *et al*, 2015). Protein intensities were $\log_2$-transformed. All the above datasets were then processed as described below.

### Data pre-processing

Every protein was required to be quantified in more than half of the replicates (or samples of the same cell state) in both compared samples in the dataset, it was discarded otherwise. Quantile normalization was applied across samples/replicates prior to the analysis with the limma (Ritchie *et al*, 2015; Phipson *et al*, 2016) approach. Replicates were averaged for every protein before applying the CNV approach, and no prior normalization was applied in this case. Datasets were quantile-normalized when comparing the performances of the CNV and limma approaches.

## Annotation

Proteins were assigned a cellular localization, whenever possible, using GO (The Gene Ontology Consortium, 2015) cellular compartment terms through the biomaRt package (Smedley *et al*, 2017). Ten compartments and their associated sub-localizations were considered in this work and defined using the following GO terms: GO:0005634 (nucleus); GO:0005737 (cytoplasm); GO:0005739 (mitochondrion); GO:0005576, GO:0031012, GO:0044421, and GO:0044420 (extracellular compartment); GO:0005777 (peroxisome); GO:0005764 (lysosome); GO:0005783 (endoplasmic reticulum); GO:0005794 (Golgi apparatus); GO:0005886 (cell membrane); GO:0031965 (nuclear membrane). Children terms of these main compartments were derived using the GOCCOFFSPRING function of the GO.db library (https://doi.org/doi:10.18129/b9.bioc.go.db). Multiple identifiers for the same protein were considered if provided. Every protein was considered during the analysis for all the cellular compartments it was matched to.

## Data modeling

The standard differential expression analysis was conducted using the limma package (Ritchie *et al*, 2015; Phipson *et al*, 2016), fitting the data into a linear model (*lmFit* function) and estimating the empirical Bayes statistics (*eBayes* function); *q*-values were estimated using fdrtool (Strimmer, 2008). The CNV approach annotates and divides the proteome into ten main cell compartments using the GO cellular component annotation (defined above). The abundances of proteins in the two compared conditions/samples were used to build linear regression models using the *lm* function (Chambers, 1992) for each compartment. The *P*-value of the linear model was evaluated, and the linear model was retained if significant ($P < 0.05$). The residuals associated with each protein were taken from the linear model of the compartment and then standardized (CNV value). The *q*-value corresponding to each CNV value was calculated with fdrtool (Strimmer, 2008).

## Data availability

The standard analysis with limma and the CNV approach can be performed through a python/R scripts package available at https://github.com/lucaparca/cnv (Code EV1, https://doi.org/10.5281/zenodo.1237046). Analyses can be customized, depending on the presence and number of replicates, setting the number of tolerated missing values per sample, and setting the organism and the type of the protein identifier (e.g., UniProt or Ensembl). Protein grouping can be automatic (compartment annotation is performed as described in the paper) or customized (in this case, protein annotation is provided by the user). A *readme* file is provided with all the instructions needed, as well as example input files [related to two of the datasets analyzed in this work (Guo *et al*, 2015; Azimifar *et al*, 2014)].

**Expanded View** for this article is available online.

## Acknowledgements

We would like to thank Dr. Bernd Klaus from the EMBL Centre for Statistical Data Analysis for critical comments on the manuscript. The work by LP, MB, and PB is supported by the European Molecular Biology Laboratory. The FLI is a member of the Leibniz Association and is financially supported by the Federal Government of Germany and the State of Thuringia.

## Author contributions

LP, MB, PB, and AO conceived the project. LP and AO designed and performed the analyses. LP wrote the manuscript with input from the other authors. AO oversaw the project.

## Conflict of interest

The authors declare that they have no conflict of interest.

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
