## [Review Process File · Molecular Systems Biology]

Quantifying compartment-associated variations of protein abundance in proteomics data

Luca Parca, Martin Beck, Peer Bork and Alessandro Ori.

Review timeline:

Submission date:	6 th December 2017
Editorial Decision:	16 th January 2018
Revision received:	30 th April 2018
Editorial Decision:	6 th June 2018
Revision received:	11 th June 2018
Accepted:	11 th June 2018

Editor: Maria Polychronidou

Transaction Report:

1st Editorial Decision

16th January 2018

Thank you again for submitting your work to Molecular Systems Biology. We have now heard back from the three referees who agreed to evaluate your study. As you will see below, the reviewers think that the study addresses a relevant topic and the proposed approach will likely be useful. They raise however a series of concerns, which we would ask you to address in a revision of the manuscript.

The reviewers' recommendations are rather clear and therefore I think that there is no need to repeat all the points listed below. One particularly important point raised by reviewer #2 refers to the need to provide quantitative measures of the method's performance. Please let me know in case you would like to discuss further any of the issues raised by the reviewers.

REFeree REPORTS

Reviewer #1:

Quantifying compartment-associated variations of protein abundance in proteomics data

Luca Parca et al. describe the presence of observed protein abundance changes in proteomics datasets, which are caused by the morphological differences between samples studied. The authors show examples of this phenomenon by investigating several sample sets containing different cell types, cells from different regions, subtypes, healthy vs diseased cells, etc. The authors then focus their study on the major organelles, nucleus, cytoplasm and mitochondria and extracellular proteins. They convincingly show that indeed the difference or alteration of the morphology has an influence on the abundance of proteins, originating from the affected organelles and show a straight forward method to correct for this, if the localization of the identified proteins is well annotated. The final example on an aging dataset nicely illustrates the changed biological insights that can be obtained

upon correction.

The manuscript is well written, the examples are clear and the topic is certainly of interest to the broader omics community involved in systems biology applications.

Comments:

1. The authors nicely illustrate their method using proteins derived from major organelles. What is missing is the sensitivity of the method, although depend on both the observed proteins and their annotation. It would be nice to know what is the limit of the methodology and what is the sensitivity for proteins specific to other organelles.
2. Some figure legends are somewhat unclear. For figure 2, the description of panel D starts before 'D)' and therefore seems to belong to C). For figure 3f, the colouring is not clearly explained. The coloured circle I assume, represents the colours in the insert (which misses labels on the x and y-axis), the bottom left box is this insert in the middle?
3. Vitellogenin-3 is highlighted in figure 3 but not described at all.
4. Several journal names are missing in the references.

Reviewer #2:

In their manuscript "Quantifying compartment-associated variations of protein abundance in proteomics data", Parca et al describe a new method to increase the accuracy of differential protein expression analysis. They propose to apply a correction to protein expression data in the form of a "compartment-normalized variation value" (CNV). In essence, this is about the conceptual experiment design: against what set of proteins is a change measured. If done against all proteins in a cell, morphological changes may overstate specific alterations. The fact that one cell line might have larger nuclei than another is already visible by microscopy. To find out how the nuclei change one must normalise against nuclear proteins and not whole cells. CNVs are intended to correct protein abundance changes against underlying bulk changes of entire subcellular compartments. They show a nice example for why this is important: the amount of the respiratory chain complex I was thought to decrease during aging, but in reality this is not specific to this complex but merely reflects a general decrease in mitochondrial content.

I think this is an interesting and well-written paper and their CNV-normalisation should not come as a surprise as they are going to the heart of data normalisation and what conclusions can be taken from experiments. As the authors show, when done wrong one arrives at false conclusions. To me it has the touch of a tutorial, but one that should be made class room reading in proteomics. I do have a number of comments:

- I'm missing a documentation of the performance improvement in numbers. The authors analyse a lot of data, but only provide a few hand-selected proof-of-principle examples. While this is nice and useful, we need more stats about the method in general in order to understand if this is generally useful. For example, across all your data, how many proteins are significantly changed with the traditional expression analysis and with CNVs? I assume CNVs will mainly sort out "false-positive" changes, i.e. those that reflect a compartment-wide expression change rather than a protein-specific one. How many previously undiscovered changes do you find after normalizing for compartment-wide changes (such as the mitochondrial important machinery in aging), is there a difference between datasets? And so on... In short, if I perform a differential protein expression analysis, how likely am I to profit from applying CNVs?
- Many, if not the majority of proteins localise to multiple subcellular compartments (see for example the map by Christoforou et al, Nature communications, 2016). How to deal with that? Can you correct fold-changes for a combination of locations? I think the authors should at least discuss this issue.
- I find the title slightly misleading, I think you should clarify that this is not just about quantifying compartment variation, but how to normalize against it
- You might want to mention the fact that organelles change in abundance between different conditions is not only a problem, but it can also be exploited for functional proteomics (for example see Kustatscher et al, Proteomics, 2016).
- Reference 18 is incomplete (no journal).
- Page 2, line 67: sentence is confusing

- On page 3 you mention enrichment in %, but in other places in fold-changes. Please be consistent. I personally prefer fold-changes.
- Figure 1b, c, d: Can you put the bar charts inside the panels?
- Figure 2 legend is confusing: What do you mean by "estimated" abundance? Panel A, clarify that fold-change refers to whole-cell on just mitochondria (I think). Panel C is not called in the main text. The sentence "Correlation between standard fold change..." (line 409) seems to belong to an earlier draft.
- Figure 3d (and S3): I find it hard to see the effect in the line graphs. Can you add bar charts or boxplots for the complexes as whole (rather than showing individual subunits), with some indication of the significance of the changes?
- Figure S1: Is this statistically significant? Please address or leave out.

Reviewer #3:

Parca et al - Jan 2018

This manuscript describes a normalization method to correct for changes in morphology in cells that may lead to skew in quantitative proteomics data sets. Briefly, proteins identified and quantified in a mass spectrometry based quantitative proteomics experiment are divided up in subcellular categories using GO annotation and then for each category a linear model is created. Within each linear model, the distance of a protein's abundance from the linear fit is calculated and the resulting compartment normalized value (CNV) is then used to determine whether this value represents a difference in abundance above and beyond the average for that compartment.

Having developed the CNV approach, the authors applied this to some existing data sets from the literature and also applied it to a study of *C. elegans* aging providing novel biological insight from within this published dataset.

The motivation for creating this approach is that many quantitative proteomics studies that use total cell lysates from samples where the response to perturbation is being investigated or differences in cell types, do not take account of gross changes in morphology. For example, if a perturbation leads to increase numbers of mitochondria in a cell, then in theory all mitochondrial proteins would appear to be increased in abundance. Whilst this is useful information, these bulk organelle changes might mask more telling increases in individual components of this organelle. To my knowledge, the proteomics community do not consider such morphological changes when carrying out shot gun proteomics analysis and quantitative investigations of the proteome which in many cases may be an important oversight. It could be argued that cell biologists employing these tools should also have microscopy data indicating morphological changes that result from the dynamic system being studied or different cell types being compared.

I consider that this manuscript addresses a very important question. It is well written and provides clear explanation of the issues and the methods employed.

I have some reservations about the approach and the conclusions drawn as follows:

1. the method relies extremely heavily on GO Cellular Compartment annotations. It is well known that many of these annotations are erroneous leading to potential inaccuracies in the linear models. Moreover, many proteins are located in multiple places (Thul et al Science 2017). The only brief explanation of how this feature was accounted for in the creation of the methods is given in the methods section...'Multiple identifiers for the same protein were considered if provided'.....

I would like to have seen the authors explore this further by explaining how proteins per subcellular component were chosen and the performance of the method if a smaller, 'gold standard' set of the proteins were used to create the linear model per compartment.

2. The cell contains many different subcellular compartments. Here the authors only consider 4 - what about the rest?

3. Situations where all proteins within a compartment are increased or decreased by wholesale changes in the morphology are of interest and shouldn't be 'normalized' out of the dataset. The authors should more clearly explain that in their system, this important information is also retained.

4. How do the authors account for situations where protein switch location upon perturbation/different cell types?

5 Is the method sensitive to changes in sub compartments? For example, if nuclear size is different between two cell types, it does not necessarily follow that all the proteins will scale accordingly. In

such situations a single linear model may not represent the data adequately. The authors should make comment on this eventuality.

6. Absolute abundance measurements as indicated by spectral counts should also give an estimate of gross morphological changes. Although some experimental designs do not give information about this, many do, where the coverage of data from a proteins can be used as an estimate of total amount. Was use of this information considered as additional data to inform the model?

7. This method is only useful in studies that look at dynamic changes in abundance in proteins, it is not useful in studies that aim to create an organelle catalogue of proteins.

Reviewer #1:

Quantifying compartment-associated variations of protein abundance in proteomics data

Luca Parca *et al.* describe the presence of observed protein abundance changes in proteomics datasets, which are caused by the morphological differences between samples studied. The authors show examples of this phenomenon by investigating several sample sets containing different cell types, cells from different regions, subtypes, healthy vs diseased cells, etc. The authors then focus their study on the major organelles, nucleus, cytoplasm and mitochondria and extracellular proteins. They convincingly show that indeed the difference or alteration of the morphology has an influence on the abundance of proteins, originating from the affected organelles and show a straight forward method to correct for this, if the localization of the identified proteins is well annotated. The final example on an aging dataset nicely illustrates the changed biological insights that can be obtained upon correction.

The manuscript is well written, the examples are clear and the topic is certainly of interest to the broader omics community involved in systems biology applications.

Comments:

1. The authors nicely illustrate their method using proteins derived from major organelles. What is missing is the sensitivity of the method, although depend on both the observed proteins and their annotation. It would be nice to know what is the limit of the methodology and what is the sensitivity for proteins specific to other organelles.

- In order to quantify the fraction of the proteome that can be analyzed by the CNV approach, we have included an analysis of proteins that can be annotated through GO cellular component terms in the seven datasets we analyzed in this work. We show that we can assign a cell compartment to 78% of the proteins on average in the considered datasets. We have included in the revised manuscript a detailed overview of the coverage of each dataset in the new Figure EV1A, which reports the percentage of proteins that have been associated to (i) the 4 main compartments initially originally considered in this study, (ii) 6 additional compartments (Golgi apparatus, endoplasmic reticulum, cell membrane, nuclear membrane, lysosome, peroxisome), (iii) to other GO cellular component terms, (iv) no cellular compartment. The 4 main compartments (nucleus, cytoplasm, mitochondrion and extracellular) cover already 96% of all the proteins annotated through GO; this percentage increases to 97% if 6 additional compartments are considered. We included these figures in the Result section (Page 3, *"On average, 78% of the proteins in the analyzed dataset could be annotated using GO cellular compartments terms (Figure EV1A). We compared abundance changes of proteins belonging to four major cellular compartments (nucleus, cytoplasm, mitochondrion, and extracellular space) which, on average among the analyzed datasets, accounted for 96% of all the annotated proteins"*). Thus, the majority of the proteins quantified in the dataset considered can be analyzed with the CNV approach. The depth of the analysis might however limited in less well-annotated species. We included a comment regarding this last point in the Discussion (Page 8,

"However, it is dependent on compartment assignment of proteins and, therefore, it relies on accurate annotation of protein localization, thus limiting its application to well-annotated model organisms."

- We added a comment, in the discussion (Page 8, *"Moreover, the reliability of the linear models built for each compartment depends on the number of proteins used to build them, therefore the statistics of models built for organelles with fewer proteins or of small size must be taken into account."*), regarding eventual models built for organelles with few proteins (e.g. because of either small cellular compartment or poor annotation), which can determine non-significant linear model due to small set of data points.
- We provide compartment-specific Fold Change density plots for other compartments with major shifts (which are in Figure EV1C-D-E) in the three datasets explored in the first paragraph of the manuscript. We also added boxplots for the 10 compartments in main Figures 1B-C-D. From this new analysis, we identified a previously unappreciated correlation between the abundance of proteins annotated as peroxisomal and mitochondrial throughout all the datasets considered (pearson R 0.97, p -value= 3.5×10^{-4}). We included a comment on this new observation in the manuscript (Page 4, *" This allowed us to observe a previously unappreciated correlation between the abundance changes of proteins annotated as peroxisomal and mitochondrial (Pearson R 0.97, $p=3 \times 10^{-04}$) that manifested in all the seven different datasets used (Figure EV1F). Proteins annotated to both mitochondria and peroxisomes were excluded for this analysis."*)

2. Some figure legends are somewhat unclear. For figure 2, the description of panel D starts before 'D)' and therefore seems to belong to C). For figure 3f, the colouring is not clearly explained. The coloured circle I assume, represents the colours in the insert (which misses labels on the x and y-axis), the bottom left box is this insert in the middle?

- We thank the reviewer for pointing this out. Corrections were made as suggested and the legend of Figure 3 has been updated (Page 17, *" Four proteins have been selected (highlighted with a colored circle: purple for NAS-34, green for Y45F10C.4, blue for Vitellogenin-1, black for Y45F10C.2) for their slower or accelerated increase of abundance compared to the rest of extracellular proteins. The variation of the selected proteins (highlighted with the same colors) throughout the worm lifespan is represented in the middle box, with the average of the extracellular proteome colored in yellow."*).

3. Vitellogenin-3 is highlighted in figure 3 but not described at all.

- We corrected a typo (Vitellogenin-1) and cited the protein, which plays an important role during the embryonic development of *C.elegans*, in the text (Page 7, *" we observed the Vitellogenin-1 (CNV value of +1.5), a precursor of egg-yolk proteins, the Y45F10C.4 protein (CNV value of +2.1)"*).

4. Several journal names are missing in the references.

- References were corrected as suggested.

Reviewer #2:

In their manuscript "Quantifying compartment-associated variations of protein abundance in proteomics data", Parca *et al.* describe a new method to increase the accuracy of differential protein expression analysis. They propose to apply a correction to protein expression data in the form of a "compartment-normalized variation value" (CNV). In essence, this is about the conceptual experiment design: against what set of proteins is a change measured. If done against all proteins in a cell, morphological changes may overstate specific alterations. The fact that one cell line might have larger nuclei than another is already visible by microscopy. To find out how the nuclei change one must normalise against nuclear proteins and not whole cells. CNVs are intended to correct protein abundance changes against underlying bulk changes of entire subcellular compartments. They show a nice example for why this is important: the amount of the respiratory chain complex I was thought to decrease during aging, but in reality this is not specific to this complex but merely reflects a general decrease in mitochondrial content.

I think this is an interesting and well-written paper and their CNV-normalisation should not come as a surprise as they are going to the heart of data normalisation and what conclusions can be taken from experiments. As the authors show, when done wrong one arrives at false conclusions. To me it has the touch of a tutorial, but one that should be made class room reading in proteomics. I do have a number of comments:

• I'm missing a documentation of the performance improvement in numbers. The authors analyse a lot of data, but only provide a few hand-selected proof-of-principle examples. While this is nice and useful, we need more stats about the method in general in order to understand if this is generally useful. For example, across all your data, how many proteins are significantly changed with the traditional expression analysis and with CNVs? I assume CNVs will mainly sort out "false-positive" changes, i.e. those that reflect a compartment-wide expression change rather than a protein-specific one. How many previously undiscovered changes do you find after normalizing for compartment-wide changes (such as the mitochondrial important machinery in aging), is there a difference between datasets? And so on... In short, if I perform a differential protein expression analysis, how likely am I to profit from applying CNVs?

- In order to assess the performance of our method across dataset, we first compared q values calculated by standard differential expression (based on limma) and our CNV approach. As shown in the new Figure EV4, q values calculated by the two approaches were only moderately correlated, indicating complementarity between the two approaches. Interestingly, the level of complementarity (as assessed by the correlation between q values) varied between dataset and it was more pronounced when distantly related tissues (i.e., liver and lung) were compared. As suggested by the reviewer, we calculated the percentage of proteins that are classified as significantly differentially regulated by the standard differential expression approach that are not significant with the

CNV approach (which would represent the “false positives” described by the reviewer) and the percentage of proteins that are significant with the CNV approach but not with the standard approach (which would represent the novel cases, undetected with the standard approach). These numbers have been calculated for the 3 datasets for which we detected strong compartment shifts (as shown in Figure 1B:D) and are reported in the new Figure 2 (panels F-G-H) with the inset barplots. On average, 66% of all the significant protein identified with the CNV approach ($q\text{-value}<0.1$) were not identified with the standard approach (colored in cyan points in the volcano plots and barplots in Figure 2-F-G-H), while a percentage, ranging from 50% to 92% (70% on average across the datasets), of the proteins identified as differentially expressed by the standard approach ($q\text{-value}<0.1$) are very close to the linear model of their respective compartment and classified as not significantly changed with the CNV approach (colored in red). It has to be noted that the linear model statistics is very strict when calling significant proteins, which is why a large percentage of previously significant proteins are classified as non-significant with the CNV approach. Despite this, a number of new cases can be identified in each dataset, and their variation relative to the respective compartment measured. We want to thank the reviewer as we now have a measure of the impact of the CNV approach on the analyzed datasets. We added this analysis to the results (Page 6, "*Comparison of proteome-wide and compartment-specific differential expression analysis. In order to quantify the impact of the CNV approach on the analysis of proteomic data, we compared the outcome of standard differential expression and CNV approach for the three dataset where we detected significant compartment biases (Figure 1-B:D). Direct comparison of the statistics revealed low correlation between q-values assigned by the two approaches (Figure EV4), indicating complementarity between them. Notably, the extent of complementarity was not uniform between datasets, being more pronounced when different tissue (e.g., liver vs. lung) are compared (Figure 2-F:H). We reasoned that the CNV can provide two additional levels of information: (i) it can reveal alterations of protein level that reflect a compartment-wide abundance change rather than a protein-specific one, and (ii) it can discover new protein changes that emerge only after normalizing for compartment-wide changes. Therefore, we explicitly investigated the overlap between significant proteins identified by standard differential expression and CNV approach. Across the three dataset tested, we found a variable proportion of cases (ranging between 50% and 92%) that were identified as differentially expressed by the standard approach ($q\text{-value}<0.1$), but are very close to the linear model of their respective compartment, and, thus, classified as not significant with the CNV approach (Figure 2-F:H, colored in red). We interpret these cases as deriving from compartment-wide abundance changes. This effect was particularly pronounced for the (Azimifar et al, 2014) dataset that showed very prominent shifts for nuclear, mitochondrial and extracellular proteins (Figure 1C). Regarding newly discovered cases, we found 104, 53 and 38 proteins that were identified as significant ($q\text{-value}<0.1$) exclusively by the CNV approach, respectively for the lung vs. liver (Geiger et al, 2013), hepatocyte vs. Kupffer cells (Azimifar et al, 2014) and healthy kidney vs. carcinoma (Guo et al, 2015) datasets (Figure 2-F:H, colored in cyan). The majority of these cases display low fold changes relatively to the total proteome, but appears as outlier in the linear models for the respective compartment. Taken together these data demonstrate*

that protein expression can be analyzed in the context of cellular compartment by building simple linear models that allow a complementary interpretation of the results of canonical differential expression (Figure EV4), revealing new differences in the abundance of proteins belonging to the same compartment across cell types and states. "

• Many, if not the majority of proteins localise to multiple subcellular compartments (see for example the map by Christoforou *et al*, Nature communications, 2016). How to deal with that? Can you correct fold-changes for a combination of locations? I think the authors should at least discuss this issue.

- We explored this issue as suggested by reviewer 2 and 3. We assessed the occurrence of proteins annotated to multiple localizations in the analyzed dataset. Indeed the majority of the proteins are annotated to more than one compartment, see Figure EV1B. We have now included a sentence describing this in text: (Page 4, "*Since many proteins are associated to more than one cellular compartment (on average only 20% of the annotated proteins were specific to one compartment and 36% were annotated to two compartments, Figure EV1B)...*").
- As outlined in the reply to reviewer 1, we performed two additional analyses in order to assess the robustness of the CNV approach when dealing with multiple localizations. First, we showed that protein annotated to multiple compartments do not influence the estimation of the linear model, and, in case of mitochondrial proteins, also tested an independent annotation (MitoCarta). We included the following text in the Result section (pages 4-5) describing this analysis: "*... we wanted to assess the robustness of the linear models when taking into account multiple compartment annotations for the same proteins. Thus, we compared the CNV models built using all the proteins mapping to a given compartment, to CNV models built using only proteins that are exclusive to a given compartment, so that there are no shared proteins between the linear models. We measured an average Pearson correlation of 0.97 between the CNV values for the same proteins using the two types of models. In the case of mitochondria, we evaluated an independent and curated annotation of mitochondrial proteins from MitoCarta (Calvo *et al*, 2016), and used it to build the mitochondrial-exclusive CNV model. The average Pearson correlation between the previous models and these mitochondrion-exclusive models was 0.99. The statistics of the CNV models and their correlation with compartment-exclusive models are reported in the Dataset EV4A. "*
- Subsequently, we assessed whether there exists a significant correlation between multiple compartment annotation for proteins and their classification as differentially regulated by the CNV approach. This is based on the hypothesis that applying a wrong linear model to a protein localizing to multiple compartments would lead to higher likelihood of detecting it as outlier (differential expressed relatively to the compartment) in our analysis. As it is reported in the new Dataset EV4B there is no significant association between these two events (Fisher exact test adjusted p -value >0.01) in any of the dataset tested. We describe this

additional analysis in Page 5 of the revised manuscript: "*Finally, we tested whether proteins belonging to different compartments are more likely to be detected as significantly affected by the CNV approach. We did not observe a significant association between multiple compartment annotation for proteins and their classification as differentially expressed by the CNV approach (Dataset EV4B). Thus, we conclude that the linear models underpinning the CNV approach are robust whether proteins with multiple compartments are considered or not.*").

• I find the title slightly misleading, I think you should clarify that this is not just about quantifying compartment variation, but how to normalize against it

- We thank the reviewer for the suggestion. After consultation with the editor, we opted to keep the original, more general, version of the title.

• You might want to mention the fact that organelles change in abundance between different conditions is not only a problem, but it can also be exploited for functional proteomics (for example see Kustatscher et al, Proteomics, 2016).

- We thank the reviewer for the suggestion, we edited the introduction accordingly (Page 2, "*Covariation of protein abundance in different conditions can be also exploited and can contribute to functional proteomics (Kustatscher et al, 2016).*").

• Reference 18 is incomplete (no journal).

- All the incomplete references have been corrected.

• Page 2, line 67: sentence is confusing

- The sentence has been rephrased to improve the readability (Page 2, "*Currently, there is a lack of systematic analysis approaches able to detect and deal with differences in cellular organization that might influence the outcome of proteomic data analysis from unfractionated samples.*")

• On page 3 you mention enrichment in %, but in other places in fold-changes. Please be consistent. I personally prefer fold-changes.

- Corrections were made as suggested.

• Figure 1b, c, d: Can you put the bar charts inside the panels?

- Corrections were made as suggested. We also added a dashed line set on fold change=0 to ease the readability of the density plots and bar charts.

• Figure 2 legend is confusing: What do you mean by "estimated" abundance? Panel A, clarify that fold-change refers to whole-cell on just mitochondria (I think). Panel C is not called in the main text. The sentence "Correlation between standard fold change..." (line 409) seems to belong to an earlier draft.

- We corrected the Figure 2 as suggested by the reviewer. The “estimated” abundance is absolute quantification of proteins through IBAQ scores, we added this detail in the legend. The fold change in the panel A refers to mitochondrial proteins and not to the whole cell; we modified the legend to make this clear. Panel C is now called in the text (page 5, “*Depending on the extent of the cellular compartment shift in whole proteome data, proteins can be assigned different standard fold change and CNV values (Figure 2C).*”). The description of panel D has been modified to avoid repetition and misunderstanding.
- **Figure 3d (and S3): I find it hard to see the effect in the line graphs. Can you add bar charts or boxplots for the complexes as whole (rather than showing individual subunits), with some indication of the significance of the changes?**
 - We added bigger boxplots representing the whole complex at each time points, these boxplots are colored depending on the time point (same coloring as in Figure 3A-B-C). Individual subunits are all colored in grey to facilitate the view of the whole complex as a whole through the boxplots. We also added a red star to those boxplots representing significant changes (Mann-Whitney adjusted $p < 0.05$).
- **Figure S1: Is this statistically significant? Please address or leave out.**
 - It is indeed significant, we added the p-values of the t-test between the two pairs of distributions in the main text (Page 3-4, “*In addition, we also observed progressive shifts in the relative abundance of nuclear (t-test $p < 2.2 \times 10^{-16}$) and extracellular (t-test $p < 2.2 \times 10^{-16}$) proteins ...*”) and in the legend of the Figure EV2 (Page 15, “*Cellular compartment shifts during colorectal cancer progression, cancer/healthy total protein ratio against cancer/adenoma total protein ratio, of respectively 3231 and 3206 nuclear proteins (t-test $p < 2.2 \times 10^{-16}$) and of respectively 2172 and 2155 extracellular proteins (t-test $p < 2.2 \times 10^{-16}$) proteins (Wiśniewski et al, 2015) across three conditions: healthy, adenoma and cancer.*”).

Reviewer #3:

Parca et al - Jan 2018

This manuscript describes a normalization method to correct for changes in morphology in cells that may lead to skew in quantitative proteomics data sets. Briefly, proteins identified and quantified in a mass spectrometry based quantitative proteomics experiment are divided up in subcellular categories using GO annotation and then for each category a linear model is created. Within each linear model, the distance of a protein's abundance from the linear fit is calculated and the resulting compartment normalized value (CNV) is then used to determine whether this value represents a difference in abundance above and beyond the average for that compartment.

Having developed the CNV approach, the authors applied this to some existing data sets from the literature and also applied it to a study of *C.elegans* aging providing novel biological insight from within this published dataset. The motivation for creating this approach is that many quantitative proteomics studies that use total cell lysates from samples where the response to perturbation is being investigated or differences in cell types, do not take account of gross changes in morphology. For example, if a perturbation leads to increase numbers of mitochondria in a cell, then in theory all mitochondrial proteins would appear to be increased in abundance. Whilst this is useful information, these bulk organelle changes might mask more telling increases in individual components of this organelle. To my knowledge, the proteomics community do not consider such morphological changes when carrying out shot gun proteomics analysis and quantitative investigations of the proteome which in many cases may be an important oversight. It could be argued that cell biologists employing these tools should also have microscopy data indicating morphological changes that result from the dynamic system being studied or different cell types being compared. I consider that this manuscript addresses a very important question. It is well written and provides clear explanation of the issues and the methods employed. I have some reservations about the approach and the conclusions drawn as follows:

1. the method relies extremely heavily on GO Cellular Compartment annotations. It is well known that many of these annotations are erroneous leading to potential inaccuracies in the linear models. Moreover, many proteins are located in multiple places (Thul et al Science 2017). The only brief explanation of how this feature was accounted for in the creation of the methods is given in the methods section...'Multiple identifiers for the same protein were considered if provided'..... I would like to have seen the authors explore this further by explaining how proteins per subcellular component were chosen and the performance of the method if a smaller, 'gold standard' set of the proteins were used to create the linear model per compartment.

- We thank the Reviewer for this comment. Indeed, the fact that proteins are annotated to multiple compartments has also been raised by reviewer #2 and has been addressed in the response above. Briefly, we explored the robustness of the CNV approach when dealing with proteins annotated with multiple compartments by comparing the CNV as described in the manuscript with compartment-exclusive proteins as suggested by the reviewer. In addition, for mitochondrial proteins we used an independently curated annotation (MitoCarta) that we consider as a “gold standard” for this compartment. In each case, we assessed the correlation between the original models (build using all proteins annotated to a compartment) and the one build using compartment-exclusive proteins or ‘gold standard’. We observed no significant difference between the resulting CNV models (average R Pearson correlation 0.97, with a minimum of 0.87 and a maximum of 0.99). This indicates that, at least for the dataset and compartments considered, the CNV approach built robust linear models regardless of proteins with multiple compartments. The complete results are in the new Dataset EV4A and EV4B. We added this analysis to the manuscript, Page 4-5: "*Since many proteins are associated to more than one cellular compartment (on average only 20% of the annotated proteins were specific to*

one compartment and 36% were annotated to two compartments, Figure EV1B) (Thul et al, 2017), we wanted to assess the robustness of the linear models when taking into account multiple compartment annotations for the same proteins. Thus, we compared the CNV models built using all the proteins mapping to a given compartment, to CNV models built using only proteins that are exclusive to a given compartment, so that there are no shared proteins between the linear models. We measured an average Pearson correlation of 0.97 between the CNV values for the same proteins using the two types of models. In the case of mitochondria, we evaluated an independent and curated annotation of mitochondrial proteins from MitoCarta (Calvo et al, 2016), and used it to build the mitochondrial-exclusive CNV model. The average Pearson correlation between the previous models and these mitochondrion-exclusive models was 0.99. The statistics of the CNV models and their correlation with compartment-exclusive models are reported in the Dataset EV4A. Finally, we tested whether proteins belonging to different compartments are more likely to be detected as significantly affected by the CNV approach. We did not observe a significant association between multiple compartment annotation for proteins and their classification as differentially expressed by the CNV approach (Dataset EV4B). Thus, we conclude that the linear models underpinning the CNV approach are robust whether proteins with multiple compartments are considered or not."

2. The cell contains many different subcellular compartments. Here the authors only consider 4 - what about the rest?

- This is a valid point that has been raised also by reviewer #1 and discussed in the response above. Briefly, we initially focused our analysis on the major four cellular compartments since, as shown in Figure EV1A, they cover already a large fraction of quantified proteins in the analyzed dataset. We have now extended our analysis to additional compartments (peroxisome, lysosome, endoplasmic reticulum, Golgi apparatus, nuclear membrane and cell membrane), and by doing so unveiled a previously unappreciated correlation between the abundance of mitochondrial and peroxisomal proteins (shown in Figure EV1F). We added the following part to the manuscript, Page 4: "*Subsequently, we extended the analysis to proteins mapping to six additional organelles: endoplasmic reticulum, Golgi apparatus, cell membrane, nuclear membrane, lysosome, and peroxisome (Figure EV1C:E). This allowed us to observe a previously unappreciated correlation between the abundance changes of proteins annotated as peroxisomal and mitochondrial (Pearson R 0.97, p=3×10⁻⁰⁴) that manifested in all the seven different datasets used (Figure EV1F).*"
- We also included a comment in the text, describing how small organelles with fewer proteins or with poor annotation can determine a small number of data points resulting in non-significant models (Page 8, "*Moreover, the reliability of the linear models built for each compartment depends on the number of proteins used to build them, therefore the statistics of models built for organelles with fewer proteins or of small size must be taken into account.*").

3. Situations where all proteins within a compartment are increased or decreased by wholesale changes in the morphology are of interest and shouldn't be 'normalized' out of the dataset. The authors should more clearly explain that in their system, this important information is also retained.

- We fully agree with the reviewer. The CNV approach is to apply “in addition” in standard differential protein expression analysis to integrate them by analyzing variations in the proteomes at different levels of proteome organization. We highlighted this important point in the discussion as suggested by the reviewer (Page 7-8, "*This results in global fold change shifts for proteins associated to different compartments. Such abundance shifts are extremely robust (deriving from tens to hundreds of proteins) and they can be used as markers of cell identity. Currently used data processing approaches do not contemplate such differences that affect collectively large portions of the proteome. This poses limitations to the detection of variations in composition of cellular compartments. The aim of the CNV approach presented here is to integrate standard analysis by revealing compartment-specific changes that can be hidden in whole proteome data.*").

4. How do the authors account for situations where protein switch location upon perturbation/different cell types?

- Detection of localization switches of proteins upon perturbation is currently not included in the CNV approach. Currently, the CNV approach will provide multiple CNV values to a protein that has been associated to multiple compartments. In principle, this could give an indication of protein translocation between cellular compartments (in case of proteins showing CNV values of opposite signs in two or more compartments), but we feel that the identification of localization changes is beyond the scope of this work and needs to be addressed using dedicated approaches.

5 Is the method sensitive to changes in sub compartments? For example, if nuclear size is different between two cell types, it does not necessarily follow that all the proteins will scale accordingly. In such situations a single linear model may not represent the data adequately. The authors should make comment on this eventuality.

- We expanded and commented a part of the discussion dedicated to this point. The CNV approach is applied to compartments, but the same concept can in theory be applied at smaller scales to *a-priori* protein clustering/grouping (e.g. protein complexes or pathway). We added a comment on this issue at the end of the discussion (Page 8, "*This would extend the reach of the analysis, revealing specific changes at a smaller scale similarly to what we have done with the CNV approach, which revealed compartment-specific changes that could not be detected with standard approaches applied to the whole cell.*")

6. Absolute abundance measurements as indicated by spectral counts should also give an estimate of gross morphological changes. Although some experimental designs do not give information about this, many do, where the coverage of data from

a proteins can be used as an estimate of total amount. Was use of this information considered as additional data to inform the model?

- Some of the analyzed datasets in this work provided relative abundance quantification for the detected proteins (i.e., SILAC quantification based on common spike-in standard in the murine lung versus liver dataset (Geiger *et al.*, 2013)), while in the majority of the analyzed datasets proteins were assigned a score related to absolute abundance based on summed peptide intensities (i.e., IBAQ scores, (Swanhäusser *et al.*, 2013)). Spectral counts were not considered in this work in favor of IBAQ as protein abundance estimates, but we envisage that they could be also used in a similar fashion using appropriate statistical models for discrete count-based data. However, we have not explored such option.

7. This method is only useful in studies that look at dynamic changes in abundance in proteins, it is not useful in studies that aim to create an organelle catalogue of proteins.

- We fully agree with this comment and added it in the final discussion (Page 8, "*However, it is dependent on compartment assignment of proteins and, therefore, it relies on accurate annotation of protein localization, thus limiting its application to well-annotated model organisms. For this reason, the CNV approach cannot be useful for the annotation of organelle catalogues of proteins.*").

2nd Editorial Decision6th June 2018

Thank you again for sending us your revised manuscript. We have now heard back from reviewer #2 who was asked to evaluate your study. As you will see below, the reviewer thinks that most issues have been satisfactorily addressed. S/he raises a few remaining issues, which we would ask you to address in a minor revision.

REFEREE REPORTS

Reviewer #2:

Parca et al have addressed all my comments appropriately. I think this is a very nice paper now.

There are only a few minor points which I missed the first time round:

- Abstract, line 35, typo: "dataset" should be "datasets"
- Results: The authors use the t-test to evaluate significance of organelle fold-changes, e.g. for the mitochondria shift in Figure 1B. However, it looks to me like many of these distributions may not be normal. Therefore, shouldn't you be using a Mann-Whitney U test instead?
- The online submission form indicates that "There is conflict of interest" but the manuscript PDF states that there is none.

2nd Revision - authors' response11th June 2018

Thank you very much for handling our manuscript. Please find enclosed a revised version of our manuscript where we addressed the remaining comments of the reviewer and your suggestions.

In the revised manuscript:

- we have used Mann-Whitney U test to assess significance of organelle fold-changes instead of t-test. This change did not affect any of our conclusions, but led to minor changes to the text (reported p-values) and Figure 1 (significance asterisks in panels 1B-D);
- we fixed the typo indicated by the reviewer in Abstract, line 35, typo: "dataset" should be "datasets";
- we fixed additional typos, modified few sentences, and improved the description of column headers in some of the supplementary tables for improved clarity.

Additionally, we have un-ticked the "Conflict of interest" box in the online form that has been erroneously ticked in a previous submission. I confirm that we have no conflict of interest as indicate in the manuscript.

All the modifications are highlighted by track changes in the submitted Word document.